# Learning Robust Kernel Ensembles with Kernel Average Pooling

## Abstract

Model ensembles have long been used in machine learning to reduce the variance in individual model predictions, making them more robust to input perturbations. Pseudo-ensemble methods like *dropout* have also been commonly used in deep learning models to improve generalization. However, the application of these techniques to improve neural networks' robustness against input perturbations remains underexplored. We introduce *Kernel Average Pooling (KAP)*, a neural network building block that applies the mean filter along the kernel dimension of the layer activation tensor. We show that ensembles of kernels with similar functionality naturally emerge in convolutional neural networks equipped with *KAP* and trained with backpropagation. Moreover, we show that when trained on inputs perturbed with additive Gaussian noise, KAP models are remarkably robust against various forms of adversarial attacks. Empirical evaluations on CIFAR10, CIFAR100, TinyImagenet, and Imagenet datasets show substantial improvements in robustness against strong adversarial attacks such as AutoAttack without training on any adversarial examples.

## 1 Introduction

Model ensembles have long been used to improve robustness in the presence of noise. Classic methods like bagging (Breiman, 1996), boosting (Freund, 1995; Freund et al., 1996), and random forests (Breiman, 2001) are established approaches for reducing the variance in estimated prediction functions that build on the idea of constructing strong predictor models by combining many weaker ones. As a result, performance of these ensemble models (especially random forests) is surprisingly robust to noise variables (i.e. features) (Hastie et al., 2009).

Model ensembling has also been applied in deep learning (Zhou et al., 2001; Agarwal et al., 2021; Liu et al., 2021; Wen et al., 2020; Horváth et al., 2022). However, the high computational cost of training multiple neural networks and averaging their outputs at test time can quickly become prohibitively expensive (also see work on averaging network weights across multiple fine-tuned versions (Wortsman et al., 2022)). To tackle these challenges, alternative approaches have been proposed to allow learning pseudo-ensembles of models by allowing individual models within the ensemble to share parameters (Bachman et al., 2014; Srivastava et al., 2014; Hinton et al., 2012; Goodfellow et al., 2013). Most notably, *dropout* (Hinton et al., 2012; Srivastava et al., 2014) was introduced to approximate the process of combining exponentially many different neural networks by "dropping out" a portion of units from layers of the neural network for each batch.

While these techniques often improve generalization for i.i.d. sample sets, they are not as effective in improving the network's robustness against input perturbations and in particular against *adversarial attacks* (Wang et al., 2018). Adversarial attacks (Szegedy et al., 2013; Biggio et al., 2013; Goodfellow et al., 2014), slight but carefully constructed input perturbations that can significantly impair the network's performance, are one of the major challenges to the reliability of modern neural networks. Despite numerous works on this topic in recent years, the problem remains largely unsolved (Kannan et al., 2018; Madry et al., 2017; Zhang et al., 2019; Sarkar et al., 2021; Pang et al., 2020; Bashivan et al., 2021; Rebuffi et al., 2021; Gowal et al., 2021). Moreover, the most effective empirical defense methods against adversarial attacks (e.g. adversarial training (Madry et al., 2017) and TRADES (Zhang et al., 2019)) are extremely computationally demanding

(although see more recent work on reducing their computational cost (Wong et al., 2019; Shafahi et al., 2019)).

Our central premise in this work is that *if ensembles can be learned at the level of features (the unit activity at the intermediate layers of the network; in contrast to class likelihoods), the resulting hierarchy of ensembles in the neural network could potentially lead to a much more robust classifier.* To this end, we propose a simple method for learning ensembles of kernels in deep neural networks that significantly improves the network's robustness against adversarial attacks. In contrast to prior methods such as *dropout* that focus on minimizing feature co-adaptation and improving the individual features' utility in the absence of others, our method focuses on learning *feature ensembles* that form local "committees" similar to those used in Boosting and Random Forests. To create these committees in layers of a neural network, we introduce the *Kernel Average Pooling (KAP)* operation that computes the average activity in nearby kernels within each layer – similar to how spatial Average Pooling layer computes the locally averaged activity within each spatial window, but instead along the kernel dimension. We show that incorporating KAP into convolutional networks leads to learning kernel ensembles that are topographically organized across the tensor dimensions over which the kernels are arranged (i.e. kernels are arranged in a vector or matrix according to their functional similarity). When such networks are trained on inputs perturbed by additive Gaussian noise, these networks demonstrate a substantial boost in robustness against adversarial attacks. In contrast to other ensemble approaches to adversarial robustness, our approach does not seek to train multiple independent neural network models and instead focuses on learning kernel ensembles within a single neural network. Moreover, compared to neural network robustness methods such as Adversarial Training (Madry et al., 2017) and TRADES (Zhang et al., 2019), training on Gaussian noise is about an order of magnitude more computationally efficient.

Our contributions are as follows:

- We introduce kernel average pooling as a simple method for learning kernel ensembles in deep neural networks.

- We demonstrate how kernel average pooling leads to learning topographically organized kernel ensembles that in turn substantially improve model robustness against input noise.

- Through extensive experiments on a wide range of benchmarks, we demonstrate the effectiveness of kernel average pooling on robustness against strong adversarial attacks.

## 2 Related works and background

**Adversarial attacks:** despite their superhuman performance in many vision tasks such as visual object recognition, neural network predictions can become highly unreliable in the presence of input perturbations, including naturally- and artificially-generated noise. While performance robustness of predictive models to natural noise has long been studied in the literature, more modern methods have been invented in the past decade to allow discovering small model-specific noise patterns (i.e. adversarial examples) that could maximize the model's risk (Szegedy et al., 2013; Biggio et al., 2013; Goodfellow et al., 2014).

Numerous adversarial attacks have been proposed in the literature during the past decade (Carlini & Wagner, 2017; Croce & Hein, 2020; Moosavi-Dezfooli et al., 2016; Andriushchenko et al., 2020; Brendel et al., 2017; Gowal et al., 2019). These attacks seek to find artificially generated samples that maximize the model's risk. Formally, given a classifier function $f_\theta : \mathcal{X} \rightarrow \mathcal{Y}, \mathcal{X} \subseteq \mathbb{R}^n, \mathcal{Y} = \{1, ..., C\}$, denote by $\pi(\mathbf{x}, \epsilon)$ a perturbation function (i.e. adversarial attack) which, for a given $(x, y) \in \mathcal{X} \times \mathcal{Y}$, generates a perturbed sample $x' \in \mathcal{B}(x, \epsilon)$ within the $\epsilon$-neighborhood of $x$, $\mathcal{B}(x, \epsilon) = \{\mathbf{x}' \in \mathcal{X} : \|x' - x\|_p < \epsilon\}$, by solving the following maximization problem

$$\max_{t \in \mathcal{B}(x,\epsilon)} \mathcal{L}(f_\theta(t), y), \tag{1}$$

where $\mathcal{L}$ is the classification loss function (i.e. classifier's risk) and $\|.\|_p$ is the $L_p$ norm function. We refer to solutions $\mathbf{x}'$ of this problem as *adversarial examples.*

**Adversarial defenses:** Concurrent to the research on adversarial attacks, numerous methods have also been proposed to defend neural networks against these attacks (Kannan et al., 2018; Madry et al., 2017;

Zhang et al., 2019; Sarkar et al., 2021; Pang et al., 2020; Bashivan et al., 2021; Robey et al., 2021; Sehwag et al., 2022; Rebuffi et al., 2021; Gowal et al., 2021). Formally, the goal of these defense methods is to guarantee that the model predictions match the true label not only over the sample set but also within the $\epsilon$-neighborhood of samples $\mathbf{x}$. Adversarial training, which is the most established defense method to date, formulates adversarial defense as a minimax optimization problem through which the classifier's risk for adversarially perturbed samples is iteratively minimized during training (Madry et al., 2017). Likewise, other prominent methods such as ALP (Kannan et al., 2018) and TRADES (Zhang et al., 2019), encourage the classifier to predict matching labels for the original ($\mathbf{x}$) and perturbed samples ($\mathbf{x}'$).

Despite the continuing progress towards robust neural networks, most adversarial defenses remain computationally demanding, requiring an order of magnitude or more computational resources compared to normal training of these networks. This issue has highlighted the dire need for computationally cheaper defense methods that are also scalable to large-scale datasets such as Imagenet. In that regard, several recent papers have proposed alternative methods for discovering diverse adversarial examples at a much lower computational cost and have been shown to perform competitively with adversarial training using costly iterative attacks like Projected Gradient Descent (PGD) (Wong et al., 2019; Shafahi et al., 2019).

Another line of work has proposed utilizing random additive noise as a way to empirically improve the neural network robustness (Liu et al., 2018; Wang et al., 2018; He et al., 2019) and to derive robustness guarantees (Cohen et al., 2019; Lecuyer et al., 2019). While, some of the proposed defenses in this category have later been discovered to remain vulnerable to other forms of attacks (Tramer et al., 2020), there is a growing body of work that highlights the close relationship between robustness against random perturbations (e.g. Gaussian noise) and adversarial robustness (Dapello et al., 2021; Ford et al., 2019; Cohen et al., 2019). Also related to our present work, (Xie et al., 2019) showed that denoising feature maps in neural networks together with adversarial training leads to large gains in robustness against adversarial examples. (Yan et al., 2021; Bai et al., 2022) showed that reweighting channel activations could help further improving the network robustness during adversarial training. However, these works are fundamentally different from our proposed method in that the focus of (Xie et al., 2019) is on *denoising* individual feature maps by considering the distribution of feature values across the spatial dimensions within each feature map, while (Yan et al., 2021; Bai et al., 2022) propose methods for regulating channel activity in the context of adversarial training.

**Ensemble methods:** Ensemble methods have long been used in machine learning and deep learning because of their effectiveness in improving generalization and obtaining robust performance against input noise (Hastie et al., 2009). In neural networks, pseudo-ensemble methods like *dropout* (Hinton et al., 2012) create and simultaneously train an ensemble of "child" models spawned from a "parent" model using parameter perturbations sampled from a perturbation distribution (Bachman et al., 2014). Through this procedure, pseudo-ensemble methods can improve generalization and robustness against input noise. Another related method is MaxOut (Goodfellow et al., 2013) which proposes an activation function that selects the maximum output amongst a series of unit outputs.

Naturally, similar ideas consisting of neural network ensembles have been tested in recent years to improve prediction variability and robustness in neural networks with various degrees of success (Pang et al., 2019; Kariyappa & Qureshi, 2019; Abbasi et al., 2020; Horváth et al., 2022; Liu et al., 2021). Defenses based on ensemble of attack models (Tramèr et al., 2018) were also previously proposed where adversarial examples were transferred from various models during adversarial training to improve the robustness of the model. Several other works have focused on enhancing the diversity among models within the ensemble with the goal of making it more difficult for adversarial examples to transfer between models (Pang et al., 2019; Kariyappa & Qureshi, 2019). However these ensemble models still remain prone to ensembles of adversarial attacks (Tramer et al., 2020).

# 3 Methods

## 3.1 Preliminaries

Let $f_\theta(\mathbf{x}) : \mathcal{X} \to \mathcal{Y}$ be a classifier with parameters $\theta$ where, $\mathcal{X} \subseteq \mathbb{R}^D$, $\mathcal{Y} = \{1, ..., C\}$. In feed-forward deep neural networks, the classifier $f_\theta$ usually consists of simpler functions $f^{(l)}(\mathbf{x})$, $l \in \{1, \ldots, L\}$ composed

together such that the network output is computed as $\hat{y} = f^{(L)}(f^{(L-1)}(\dots f^{(1)}(\mathbf{x})))$. For our function $f_\theta$ to correctly classify the input $\mathbf{x}$, we wish for it to attain a small risk for $(\mathbf{x}, y) \sim \mathcal{D}$ as measured by loss function $\mathcal{L}$. Additionally, for our classifier to be robust, we also wish $f_\theta$ to attain a small risk in the vicinity of all $\mathbf{x} \in \mathcal{X}$, normally defined by a $p$-norm ball of fixed radius $\epsilon$ around the sample points (Madry et al., 2017).

Intuitively, a model which has a high prediction variance (or similarly high risk variance) to noisy inputs, is more likely to exhibit extreme high risks for data points sampled from the same distribution (i.e. adversarial examples). Indeed, classifiers that generate lower variance predictions are often expected to generalize better and be more robust to input noise. For example, classic ensemble methods like *bagging*, *boosting*, and *random forests* operate by combining the decisions of many weak (i.e. high variance) classifiers into a stronger one with reduced prediction variance and improved generalization performance (Hastie et al., 2009).

Given an ensemble of predictor functions $f_i, i \in 1, \dots, K$ with zero or small biases, the ensemble prediction (normally considered as the mean prediction $\bar{y} = \frac{1}{K}\sum_{i=1}^{K} \hat{y}_i$) reduces the expected generalization loss by shrinking the prediction variance. To demonstrate the point, one can consider $K$ i.i.d. random variables with variance $\sigma^2$ and their average value that has a variance of $\frac{\sigma^2}{K}$. Based on this logic, one can expect ensembles of neural network classifiers to be more robust in the presence of noise or input perturbations in general. However, several prior ensemble models have been shown to remain prone to ensembles of adversarial attacks with large epsilons (Tramer et al., 2020). One reason for the ensemble models to remain vulnerable to adversarial attacks is that individual networks participating in these ensembles may still learn different sets of non-robust representations leaving room for the attackers to find common weak spots across all individual models within the ensemble. Additionally, while larger ensembles may be effective in that regard, constructing ever-larger ensemble classifiers might quickly become infeasible, especially in the case of neural network classifiers.

One possible solution could be to focus on learning robust features by forming ensembles of features in the network. Indeed, learning robust features has been suggested as a way towards robust classification (Bashivan et al., 2021). Consequently, if individual kernels within a single network are made robust through ensembling, it would become much more difficult to find adversaries that can fool the full network. In the next section, we introduce *Kernel Average Pooling* for learning ensembles of kernels with better robustness properties against input perturbations.

### 3.2 Kernel average pooling (KAP)

Mean filters (a.k.a., average pooling) are widely accepted as simple noise suppression mechanisms in computer vision. For example, spatial average pooling layers are commonly used in modern deep neural networks (Zoph et al., 2018) by applying a mean filter along the spatial dimensions of the input to reduce the effect of spatially distributed noise (e.g. adjacent pixels in an image).

Here, we wish to substitute each kernel in the neural network model with an ensemble of kernels performing the same function such that the ensemble output is the average of individual kernel outputs. This can be conveniently carried out by applying the average pooling operation along the kernel dimension of the input tensor.

Given an input $\boldsymbol{z} = [z_1, \dots, z_{N_k}] \in \mathbb{R}^{N_k}$, the kernel average pooling operation (KAP) with kernel size $K$ and stride $S$, computes the function

$$\bar{z}_i = \frac{1}{K} \sum_{l=Si-\frac{K-1}{2}}^{Si+\frac{K-1}{2}} z_l \tag{2}$$

Where $z_l$ is zero-padded (with zero weight in the computation of the average) to match the dimensionality of $\bar{\boldsymbol{z}}$ and $\boldsymbol{z}$ variables (see A.1 for the details of padding). Importantly, when $\boldsymbol{z}$ is the output of an operation linear with respect to the weights on an input $x$ (e.g. linear layers or convolutional layers), KAP is functionally equal to computing the locally averaged weights within the layer and could be interpreted as a form of Kernel Smoothing (Wang et al., 2020) when the nearby kernels are more or less similar to each other.

$$\bar{z}_i = \frac{1}{K} \sum_{l=Si-\frac{K-1}{2}}^{Si+\frac{K-1}{2}} \boldsymbol{w}_l \boldsymbol{x} = \left( \frac{1}{K} \sum_{l=Si-\frac{K-1}{2}}^{Si+\frac{K-1}{2}} \boldsymbol{w}_l \right) \boldsymbol{x} \tag{3}$$

Moreover, the degree of overlap (i.e. parameter sharing) across kernel ensembles can be flexibly controlled by adjusting the KAP stride. Choosing stride $S = K$, produces independent kernel ensembles (no parameter sharing between ensembles), while having stride $S < K$ enforces parameter sharing across kernel ensembles.

Eq. 2 assumes that kernels are arranged along one tensor dimension. However, KAP could more generally be applied on any D-dimensional tensor arrangement of kernels. For example to apply a 2-dimensional KAP (mainly considered in our experiments) on the input $z$, one can first reshape the channel dimension of size $D$ into a 2-dimensional array of dimensions $\sqrt{D} \times \sqrt{D}$ and then apply KAP along the two kernel dimensions (see Fig. 1 for a graphical illustration and §A.1 and Alg.1 in the appendix for more details). Importantly, using higher dimensional tensor arrangements of kernels when applying KAP, allows parameter sharing across a larger number of kernel ensembles.

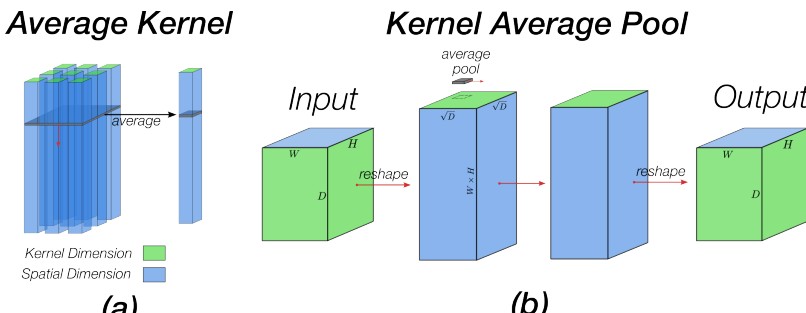

(a)          (b)

Figure 1: Graphic illustration of 2-dimensional KAP. a) the average kernel is applied per spatial position (i.e. a pixel) and computes the average of nearby kernel values at that spatial position; b) the input tensor is first reshaped such that the spatial dimensions are collapsed onto a single dimension (i.e. vectorized) and the kernel dimension is rearranged into a matrix. A 2D average pooling is applied on the reshaped tensor, and the resulting tensor is reshaped back into its original shape. Zero-padding with appropriate size is applied to maintain the original tensor shape.

In the next two subsections, we will explain how training networks with KAP-layers leads to learning topographically organized kernel ensembles (§3.3) and how these kernel ensembles may contribute to model robustness (§3.4).

## 3.3 Kernel average pooling yields topographically organized kernel ensembles

Consider a simple neural network with one hidden layer and $N_k$ units (Fig.2-left) specified by two sets of weights $\boldsymbol{W}^{(1)} = [\boldsymbol{w}_1^{(1)}, \ldots, \boldsymbol{w}_{N_k}^{(1)}]$ and $\boldsymbol{w}^{(2)} = [w_1^{(2)}, \ldots, w_{N_k}^{(2)}]$, where the hidden unit activation is $z_i = \boldsymbol{w}_i^{(1)}\boldsymbol{x}$, and the network output $y$ is computed as

$$y = \boldsymbol{w}^{(2)}\boldsymbol{z} = \sum_{i=1}^{N_k} w_i^{(2)} z_i \qquad (4)$$

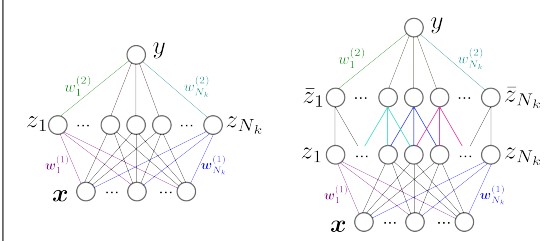

Figure 2: Schematic of a neural network with one hidden layer with (right) and without (left) the kernel average pooling operation.

In this network, the output gradients with respect to weight parameters can be computed as

$$\frac{\partial y}{\partial \boldsymbol{w}_i^{(1)}} = w_i^{(2)}\boldsymbol{x}, \qquad \frac{\partial y}{\partial w_i^{(2)}} = \boldsymbol{w}_i^{(1)}\boldsymbol{x} \qquad (5)$$

Now consider a variation of this network where the hidden unit activations are passed through a kernel average pooling layer with kernel size $K$ (Fig.2-right). In the KAP-network, the output gradients with respect to weight parameters are altered such that

$$\frac{\partial y}{\partial \boldsymbol{w}_i^{(1)}} = \frac{1}{K} \sum_{l=i-\frac{K-1}{2}}^{i+\frac{K-1}{2}} w_l^{(2)} \boldsymbol{x}, \qquad \frac{\partial y}{\partial w_i^{(2)}} = \frac{1}{K} \sum_{l=i-\frac{K-1}{2}}^{i+\frac{K-1}{2}} \boldsymbol{w}_l^{(1)} \boldsymbol{x} \tag{6}$$

where, to simplify the limits, $K$ is assumed to be an odd number. In contrast to the regular network, in the KAP-network, the gradients of the output with respect to the incoming $(\boldsymbol{w}_i^{(1)})$ and outgoing $(w_i^{(2)})$ weights in node $i$ depend on the average of outgoing and incoming weights, respectively, over the kernel average pooling window. The difference in the output gradients with respect to weights $\boldsymbol{w}_i^{(1)}$ (and similarly for $w_i^{(2)}$) for nodes $i$ and $j = i + d$ can be written as

$$\frac{\partial y}{\partial \boldsymbol{w}_i^{(1)}} - \frac{\partial y}{\partial \boldsymbol{w}_j^{(1)}} = \sum_{l=i-\frac{K-1}{2}}^{i+\frac{K-1}{2}} w_l^{(2)} x - \sum_{l=i+d-\frac{K-1}{2}}^{i+d+\frac{K-1}{2}} w_l^{(2)} x \tag{7}$$

From Eq.7, it is clear that when $K$ is large $K \gg 1$, the difference in the output gradients with respect to weights for a pair of nodes $(i, j)$ depends on the absolute difference between node indices $(|i - j|)$ and is smaller for a pair of nodes with smaller index difference. Thus, when training with backpropagation, weights connected to nodes that are closer together (in terms of their index numbers) will likely receive more similar gradients compared to those that are farther. Consequently, these weights are more likely to converge to more similar values. On the other hand, kernel sharing between ensembles provides a natural mechanism preventing the participating kernels in each group from converging to the exact same parameter values. For example, in a KAP with stride $S$ and kernel size $K$, each kernel participates in $\lceil \frac{K}{S} \rceil^2$ ensembles. Our empirical results show that the interaction between these two forces leads to smoothly transitioning topographically organized kernel maps (e.g. see Fig.7).

### 3.4 Kernel average pooling and adversarial robustness

**Relation to kernel smoothing.** Recent work (Wang et al., 2020) has suggested that the smoothness of learned kernels is one of the prominent differences between robust and non-robust models. They observed that kernels in adversarially robust models had less "dramatic fluctuations between adjacent weights" and showed that neural network models which their weights are encouraged to be smooth during training yield improved adversarial robustness. We reasoned that another approach for learning smooth kernels is to compute the average filter within an ensemble of non-smooth kernels that are tuned to identify the same feature. The biggest challenge in accomplishing this task is to learn ensembles of kernels with similar functionality or at least to be able to cluster kernels with similar functionality during training. As shown in §3.3, incorporating kernel average pooling layers in neural network models results in automatic grouping of kernels into clusters (i.e. kernel ensembles) during normal training procedure with SGD algorithm. Moreover, the average pooling operation itself provides each layer of computation with ensemble-averaged activations from previous layer, which as shown in Eq. 3 can be interpreted as a form of Kernel Smoothing.

**Relation to randomized smoothing.** Our approach is also closely related to randomized smoothing (Lecuyer et al., 2019; Cohen et al., 2019), which for a sample $(\boldsymbol{x}, y)$ consists in learning to predict $y$ with higher probability than other classes over $\mathcal{N}(\boldsymbol{x}, \sigma^2 \boldsymbol{I})$. When combined with noise resampling (running many noise-corrupted copies of the input) to estimate the true class, randomized smoothing yields certifiable robustness against adversarial perturbations. More recently, Horváth et al. (2022) show that in addition to injecting noise in the input, averaging the logits over an ensemble of models leads to a reduction of the variance due to the noisy inputs in randomized smoothing and in turn improves the certified robustness radius. They propose to learn an ensemble

$$g(\boldsymbol{x}) = Ens(f_c(\boldsymbol{x} + n)) = \frac{1}{L} \sum_l f_c^{(l)}(\boldsymbol{x} + n) \tag{8}$$

where $n \sim \mathcal{N}(0, \sigma^2 \boldsymbol{I})$ and $f_c^{(l)}$ denotes the $l$-th presoftmax classifier in an ensemble of $L$ classifiers.

As in randomized smoothing (Lecuyer et al., 2019; Cohen et al., 2019), by introducing stochasticity in the input during training, we hope to learn a model that is robust to perturbations. Moreover, in a simple setting where only one layer of KAP is used without activations, we note that a KAP block (consists of additive Gaussian noise, followed by a convolution and a KAP operation) averages the features obtained from multiple models, each corresponding to a filter (see KAP block in Fig. 3). Unlike Horváth et al. (2022), our ensembling is done at the level of features instead of the logits. Nevertheless, their arguments still apply to the case of a reduction of variance in the features (instead of logits) computed. Therefore, if KAP features are used as inputs to fully connected layers to compute logits, a reduction of variance in the features will translate into a reduction of variance in the logits. An additional key difference is that in our case, in the full KAP-based models used in §4, we use networks consisting of multiple cascaded KAP blocks to extract features, similar to how multiple convolutional blocks are used sequentially in convolutional networks. Each KAP block consists in an ensembling, from an input that was perturbed with Gaussian noise. In other words, our approach can be understood as recursively performing a form of Horváth et al. (2022)'s randomized smoothing with ensembles, by interpreting the output of each KAP block as a randomized smoothing input for the next block. In this light, our KAP architectures perform the operation

$$f_c \circ g_{N_l} \circ ... \circ g_1(\boldsymbol{x}) \text{ where } g_i(\boldsymbol{x}) = Ens[f_i(\boldsymbol{x} + \boldsymbol{n}_i)] \tag{9}$$

where the ensembling is performed by the KAP operation, potentially including activation functions, $N_l$ is the number of KAP blocks, $\boldsymbol{n}_i$ is Gaussian noise injected in the input of KAP block $i$, $f_i$ is the operation (e.g. a convolution) performed in KAP block $i$ before a kernel average pooling, and $f_c$ maps the features to the logits of the classes (Fig. 3).

Our reasoning for this recursive approach is twofold: first, during training, adding noise at each layer encourages all layers to learn to be robust to variance in their input. Otherwise in deep networks, some layers may not be exposed to significant variance in their input depending on their depth, due to the variance reduction occurring in the earlier layers. Second, the ensembling at every layer further reduces the variance in feature activations and ultimately the model predictions. If this approach successfully performs randomized smoothing on the features, we may hope that this will lead the representations of adversarially perturbed inputs to remain close to the distribution of inputs perturbed with Gaussian noise, on which the network has been trained and therefore should perform well. We empirically verify this intuition in the appendix (Fig.A2,A3).

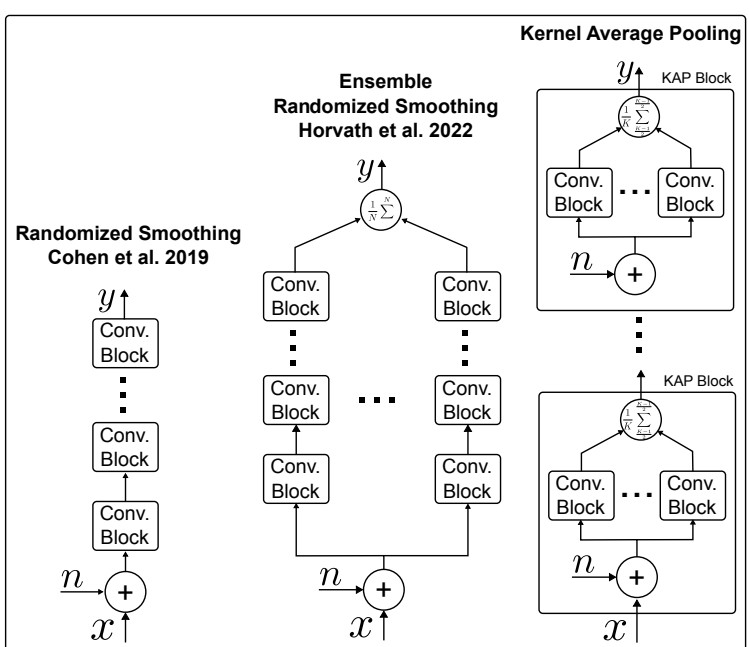

Figure 3: Schematic comparison of the architectures in (Cohen et al., 2019), (Horváth et al., 2022), and Kernel Average Pooling.

## 4 Results

In this section we empirically assess how kernel average pooling affects robustness in neural networks against adversarial attacks. We first carry out a set of experiments to invesigate the effect of different hyperparameters on the resulting model robustness (§4.2) and then inspect KAP's scaling properties by incorporating KAP into a deep convolutional network and testing its robustness properties on various computer vision benchmarks at different scales (§4.3).

### 4.1 Experimental setup

**Datasets:** We validated our proposed method on several standard benchmarks: CIFAR10, CIFAR100 (Krizhevsky, 2009), TinyImagenet (Le & Yang, 2015), and Imagenet (Deng et al., 2009). We used standard preprocessing of images consisting of random crop and horizontal flipping for all datasets. We used images of size 32 in our experiments on CIFAR datasets, 128 for TinyImagenet, and 224 for Imagenet.

**Adversarial attacks:** We assessed the model robustness against various adversarial attacks including Projected Gradient Descent (PGD) (Madry et al., 2017), SQUARE black-box attack (Andriushchenko et al., 2020) and the AutoAttack (Croce & Hein, 2020). For all attacks we used the $L_2$ norm to constrain the perturbations. Notably, AutoAttack is an ensemble attack consisting of four independent attacks and is considered as one of the strongest adversarial attacks in the field. We used $\epsilon$ value of 1.0 for experiments on all datasets. See supplementary Table A1 for full details of attacks used for model evaluation. Importantly, no input or activation noise was used during the model evaluations to prevent activation noise from potentially masking the gradients in the network.

**Training and evaluation considerations:** We used simple convolutional networks with 1-3 convolutions in our experiments in §4.2. Each convolutional layer was followed by a BatchNormalization layer (Ioffe & Szegedy, 2015) and ReLU activation. KAP layers were always added immediately after the convolution layer. In experiments in §4.3, we primarily used the ResNet18 architecture (He et al., 2016) that consists of four groups of layers where each group contains two basic residual blocks. For the KAP variations of ResNet18, we added the KAP operation after each convolution in the network. In variations of the model with activation noise (with $\sigma > 0$), we added random noise sampled from the Gaussian distribution $\mathcal{N}(0, \sigma^2)$ to the input images and after each KAP operation (and after each convolution in the original model architecture) during model training. The input and activation noise were only used during model training and all attacks were performed on the models without input or activation noise.

For adversarial training of the baseline **AT-ES** models on CIFAR10, CIFAR100, and TinyImagenet datasets, we used the normal adversarial training procedure with early stopping and $L_2$ PGD attack (Madry et al., 2017; Rice et al., 2020). We used 20 iterations for CIFAR training runs and 10 iterations for TinyImagenet. On Imagenet dataset, we used Fast-AT method (Wong et al., 2019) with the default training parameters consisting of three training phases with increasing image resolution.

**Baseline models:** To contextualize the improvements in robustness as a result of applying KAP in §4.3, we consider several baselines including (i) the original ResNet18 (i.e. without additional KAPs) trained normally (marked with **NT**); (ii) the original ResNet18 with additive input and activation noise (marked with **NT($\sigma = .$)**); (iii) the original ResNet18 trained using adversarial training with early stopping as in (Rice et al., 2020) (marked with **AT-ES**).

### 4.2 Effect of network depth, kernel ensemble size, and noise on network robustness

We first examined the effect of KAP on robustness in relatively shallow convolutional neural networks. In particular we investigated the effect of network depth, KAP kernel size, and the amount of noise used during training on the resulting network robustness.

**Effect of depth**: The theoretical discussions in §3.4 predict an improvement in robustness with increasing number of KAP layers in the network. To empirically test this prediction, we trained convolutional networks of varying depth (1-3) and compared the robustness in these networks with similar networks where a KAP layer was added after each convolution. We found that increasing number of convolutional layers as well as number of KAP layers both contributed to improved robustness against adversarial robustness (Fig. 4a).

**Effect of kernel ensemble size**: The theoretical intuitions in §3.4 suggest that larger kernel ensembles would yield greater robustness to perturbations. As shown in §3.3 the ensemble size could be controlled by the size of the KAP kernel ($K$). To test this empirically, we considered a three layer convolutional network with non-overlapping KAP layers ($S = K$) after each convolutional layer. We then varied the KAP kernel size $K$ while keeping the total number of kernel ensembles by adjusting the layer width (i.e. number of kernels) accordingly. We found that increasing the KAP kernel size (and consequently the kernel ensemble

size) improves the network robustness against AutoAttack (Fig. 4b). The resulting kernel ensembles in each of these models are shown in Fig. A1.

**Effect of Noise**: Prior work has highlighted the close link between robustness to noise and adversarial perturbations (Ford et al., 2019). Here we also examined the relationship between noise variance used during training on model's robustness to adversarial perturbations. For this, we trained a three-layer convolutional network with KAP layer after every convolution on CIFAR10 dataset where every sample was distorted by additive Gaussian noise. We then varied the standard deviation of the Gaussian noise between 0-0.2 and evaluated each model against AutoAttack-L2 (Fig. 4c). We found that training on samples with larger noise variance led to greater robustness to adversarial attacks at the cost of lower accuracy on unperturbed images.

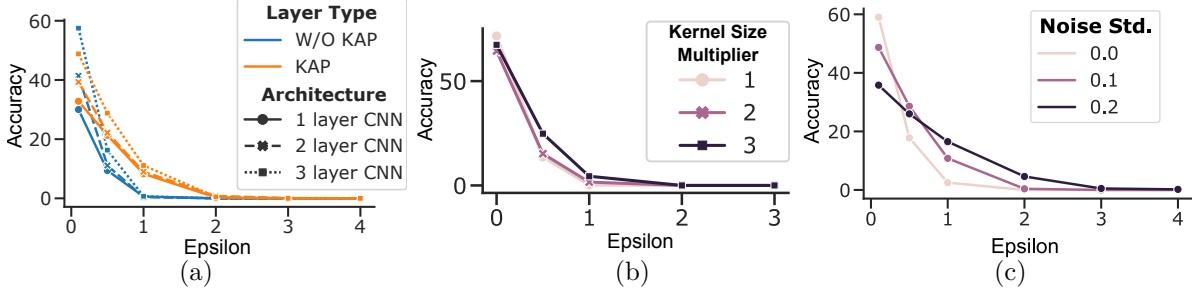

Figure 4: Effect of network depth, kernel ensemble size, and noise magnitude on robustness against AutoAttack-$L_2$ ($\epsilon = 1.0$) on CIFAR10 dataset. a) increasing the number of layers in a CNN from 1-3 significantly improves its robustness to AutoAttack in networks with KAP compared to networks without KAP; b) increasing the kernel ensemble size by increasing the KAP kernel size (Kernel Size Multiplier) improves the network robust accuracy against AutoAttack; c) Training on stronger additive Gaussian noise improves robustness to attacks with larger $L_2$ norm.

These experiments empirically confirm that adding KAP layers to convolutional networks leads to substantial improvement in accuracy under adversarial perturbations. In the next section, we extend our evaluations to deeper convolutional networks and larger datasets to investigate the scalability of using KAPs to improve robustness in neural networks.

### 4.3 Robustness in deeper networks and on larger datasets

**Robustness against adversarial attacks on CIFAR datasets.** We first compared robustness in convolutional neural networks with and without KAP on the CIFAR10 and CIFAR100 datasets. For this we trained a ResNet18 architecture and a variation of it where each convolutional layer is followed by a KAP layer ($K = 3, S = 1$).

Table 1 lists the robustness of different models trained on CIFAR10 and CIFAR100 datasets against several commonly used adversarial attacks. Confirming prior work (He et al., 2019), we found that on both CIFAR10 and CIFAR100 datasets, training the network with noisy activations can improve the network robustness. This improvement was most noticeable against the SQUARE black-box attack and to a lesser degree against PGD and AutoAttack. Furthermore, we found that adding KAP to the model significantly improves the robustness against all attacks. The amount of improvement was larger for networks trained on larger noise (i.e. larger $\sigma$), however, the higher robustness came at the cost of lower clean accuracy. Despite this, it is important to note that KAP models were trained using normal training procedures (i.e., no adversarial training) and as a result the computational cost of training was only a fraction of that required for adversarial training ($\sim 14\%$, see the supplementary Table A2 for a comparison of training speed between KAP models and adversarial training). In addition, robustness remained high against adversarial examples that were transferred from RN18-NT and RN18-AT-ES models (TableA3).

We additionally explored the effect of kernel pooling type (No pooling, Max pooling, and Average pooling) and activation noise variance ($\sigma \in \{0, 0.1, 0.2\}$) on the robust accuracy with the ResNet18 architecture. To get a better sense of how robustness generalizes to higher-strength attacks (i.e. larger $\epsilon$), we also tested each

Table 1: Comparison of adversarial accuracy against various attacks on CIFAR10 and CIFAR100 datasets. For all attacks we used an $L_2$ norm with $\epsilon = 1.0$ to constrain the adversarial examples. For each attack, we report the (mean)±(std.) of accuracy across three separate tests.

| Dataset | Model | Clean | PGD | AutoAttack | SQUARE |
|---|---|---|---|---|---|
| CIFAR10 | RN18-NT | 94.89 | 0.0 | 0.0 | 2.26±0.07 |
| | RN18-NT ($\sigma = 0.1$) | 88.44 | 9.51±0.46 | 7.05±0.29 | 34.64±0.50 |
| | RN18-AT-ES (Rice et al., 2020) | 80.95 | 38.14±0.39 | 36.75±0.91 | 56.02±1.02 |
| | TRADES (Zhang et al., 2019) | 81.79 | 52.29±1.30 | **49.64±1.07** | **63.93±0.32** |
| | RN18-KAP-NT ($\sigma = 0.1, K = 3$) | 80.00 | **63.07±0.46** | 19.44±0.59 | 20.1±0.16 |
| | RN18-KAP-NT ($\sigma = 0.2, K = 3$) | 72.21 | 59.90±0.21 | 31.28±1.19 | 32.24±1.14 |
| CIFAR100 | RN18-NT | 74.30 | 0.0 | 0.0 | 0.61±0.06 |
| | RN18-NT ($\sigma = 0.1$) | 60.05 | 5.24±0.24 | 4.80±0.12 | 19.38±0.17 |
| | RN18-AT-ES (Rice et al., 2020) | 52.31 | 16.97±0.33 | 16.12±0.73 | 28.25±0.26 |
| | TRADES (Zhang et al., 2019) | 54.17 | 27.24±0.51 | **25.06±0.35** | **35.51±0.80** |
| | RN18-KAP-NT ($\sigma = 0.1, K = 3$) | 45.70 | **31.30±1.14** | 8.46±0.21 | 9.0±0.39 |
| | RN18-KAP-NT ($\sigma = 0.2, K = 3$) | 37.30 | 30.25±0.24 | 11.74±0.48 | 12.4±0.23 |

model on a range of $\epsilon$ values (between 0.1 - 4) using AutoAttack-$L_2$ (Fig. 5). We found that **a)** adding KAP to RN18 without training the network with any activation noise already makes the network substantially more robust against attacks with smaller epsilons (Fig. 5a) and this improvement does not result from using a Kernel Max Pooling (KMP) layer; **b)** KAP model showed substantially stronger robustness to adversarial attacks compared to models without KAP (Fig. 5b), while the variation with Kernel Max Pool was the least robust variation; **c)** larger activation noise variance during training led to higher robustness against stronger attacks (Fig. 5c).

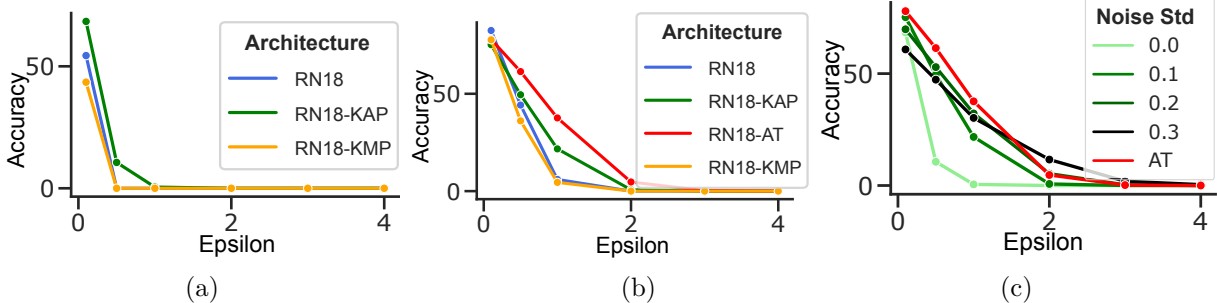

Figure 5: Robust accuracy in RN18-KAP model against AutoAttack-$L_2$ with various attack strength $\epsilon$ on CIFAR10 dataset. (a) RN18, RN18-KAP and RN18-KMP with ($\sigma = 0$, $K = 3$, $S = 1$); (b) RN18, RN18-KAP and RN18-KMP (Kernel Max Pooling) with ($\sigma = 0.1$, $K = 3$) and AT-ES; (c) RN18-KAP ($K = 3$, $S = 1$) for various noise levels $\sigma$.

In addition to improving robust classification, we also examined how the latent representations across the layers of the RN18 model with and without KAP layers were affected by the adversarial perturbations. We observed that **1)** the adversarial perturbations resulted in substantially smaller changes in the layer activations in RN18-NT-KAP model compared to RN18-NT model (Fig.A4); **2)** in RN18-NT-KAP (unlike RN18-NT model), the distribution of observed activity under adversarial perturbation was largely overlapping with that obtained for images perturbed with Gaussian noise (Fig. A2); **3)** in RN18-NT-KAP (unlike RN18-NT model), the magnitude of change in layer activations (measured by the $L_2$ norm of the differences) due to Gaussian noise and adversarial perturbations were largely similar (Fig. A3). These three observations led us to believe that compared to RN18-NT model, in RN18-NT-KAP model, the adversarial perturbations were less successful at changing the layer activation beyond that which was observed for additive Gaussian noise.

**Sensitivity of adversarial examples to noise.** Recent work has highlighted that some adversarial examples are themselves sensitive to perturbations such as Gaussian noise (Wu et al., 2021). As a result, classifiers can correctly classify these examples when small random perturbations are added to them. In addition, the experiments in previous section showed that the layer activations in the RN18-NT-KAP models were largely overlapping for adversarial and random Gaussian perturbations (Fig. A2,A3). So we asked how sensitive are the adversarial examples found for the RN18-NT-KAP model? To address this question, we investigated whether the discovered adversarial examples themselves could be rendered ineffective by adding random Gaussian noise to the input or the unit activations at inference time. We found that in all three models (RN18-NT, RN18-AT-ES and RN18-NT-KAP), many adversarial examples could be correctly classified by each network when random Gaussian noise was added to those examples at inference time. This improvement in robust accuracy was substantially higher for the RN18-NT-KAP model ($\sim$45% improvement) and for the additive input noise compared to activation noise (Table 2).

**Certified robustness:** As outlined in §3.4, our method is closely related to randomized smoothing literature and we thus expected KAP to enable stronger certified robustness too. To investigate this, we tested the RN18 architecture with and without KAP layers on their certified robustness using the standard procedure from (Cohen et al., 2019). We found that the RN18-NT-KAP model improves the certified robustness against larger noise variances (see Fig. 6) compared to RN18-NT model.

**Robustness against adversarial attacks on Imagenet.** To test whether our results could scale to larger datasets,

Table 2: Sensitivity of adversarial attacks to additive Gaussian noise. For each model the adversarial examples were generated by applying AutoAttack-$L_2$ with $\epsilon = 1$ on 1000 random CIFAR10 images. The accuracy was then tested when Gaussian noise was added to the input, intermediate layers, or both.

| Model | Input Noise | Activation Noise | AutoAttack |
|---|---|---|---|
| RN18-NT ($\sigma = 0.1$) | ✗ | ✗ | 6.10 |
| | ✗ | ✔ | 11.20 |
| | ✔ | ✗ | 24.4 |
| | ✔ | ✔ | 27.1 |
| RN18-KAP-NT ($\sigma = 0.1, K = 3$) | ✗ | ✗ | 21.70 |
| | ✗ | ✔ | 47.80 |
| | ✔ | ✗ | 67.30 |
| | ✔ | ✔ | 67.40 |
| RN18-AT-ES | ✗ | ✗ | 37.60 |
| | ✗ | ✔ | 37.60 |
| | ✔ | ✗ | 50.60 |
| | ✔ | ✔ | 50.40 |

we also trained and compared convolutional neural networks on two large-scale datasets, namely TinyImagenet (200 classes) and Imagenet (1000 classes). Here again, we used the ResNet18 architecture as our baseline and created variations of this architecture by adding KAP after every convolution operation. We found that reducing the weight decay parameter when training the KAP networks on Imagenet improves their performance and so we used a weight decay of $1e^{-5}$ in training our best models on Imagenet. In addition to the PGD-$L2$ attack, we also evaluated the models using AutoAttack on these datasets. However, because of its high computational cost, we used 1000 random samples from the validation set.

Table 3 summarizes the robust accuracy of different models on these two datasets. We found that on TinyImagenet dataset, the robustness in the KAP variation of ResNet18 was significantly higher than the

Table 3: Comparison of robust accuracy against various attacks on TinyImagenet and Imagenet datasets. For all attacks we used $\epsilon = 1$. †: models trained using Fast Adversarial Training (Wong et al., 2019). For each attack, we report the (mean)±(std.) of accuracy across three separate tests.

| Dataset | Model | Clean | PGD-$L_\infty$ | AutoAttack | SQUARE |
|---|---|---|---|---|---|
| TINYIMAGENET | RN18-NT | 57.36 | 3.63±0.45 | 3.0±0.65 | 29.60±2.56 |
| | RN18-NT ($\sigma = 0.1$) | 57.80 | 4.36±0.05 | 4.73±0.76 | 29.86±1.98 |
| | RN18-AT-ES (RICE ET AL., 2020) | 54.00 | **30.03±0.83** | **29.53±1.53** | **44.40±0.7** |
| | RN18-KAP-NT ($\sigma = 0.1, K = 3$) | 33.60 | **30.16±1.62** | 16.16±1.67 | 17.13±0.21 |
| IMAGENET | RN18-NT | 69.80 | 0.13±0.11 | 0.0 | **37.40±1.21** |
| | RN18-NT ($\sigma = 0.1$) | 48.40 | 6.03±0.35 | 4.90±1.22 | 33.93±0.97 |
| | RN18-AT† | 51.06 | 7.33±0.49 | 7.23±1.50 | 6.70±1.04 |
| | RN18-KAP-NT ($\sigma = 0.1, K = 3$) | 12.10 | 9.23±1.53 | 4.83±0.93 | 5.16±0.35 |
| | RN18WIDEX4-AT† | 63.76 | 15.00±0.31 | 15.46±2.01 | 16.40±0.78 |
| | RN18WIDEX4-KAP-NT ($\sigma = 0.1, K = 3$) | 37.40 | **34.93±1.60** | **26.73±2.31** | 27.71±0.45 |

original network and baseline trained with noise but it was still lower than the adversarially trained network. Similar to our CIFAR experiments, here we also observed that the improvements in robustness come at the cost of reduced accuracy on clean inputs.

On Imagenet, we found that the KAP variation of ResNet18 model was struggling to learn and only reached about 10-12% accuracy on the clean dataset. We reasoned that this issue could potentially be due to the large number of classes in this dataset and the possibility that the ResNet18 model does not have sufficient capacity to learn enough kernel ensembles to tackle this dataset. For this reason, we also trained a wider version of RN18 in which we multiplied the number of kernels in each layer by a factor of 4 (dubbed **RN18WideX4**). We found that this wider network significantly boosted the robust accuracy on Imagenet, even surpassing the adversarially trained model. However, we observed again that this boost to adversarial robustness was accompanied by a significant decrease in clean accuracy.

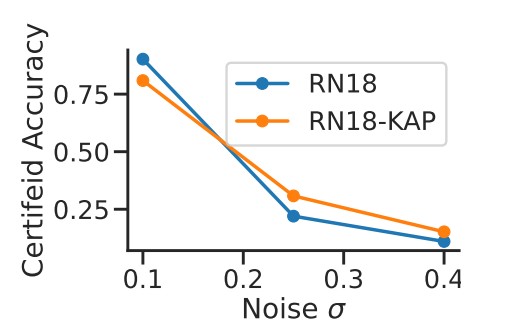

Figure 6: Certified accuracy of RN18-NT ($\sigma = 0.1$) and RN18-NT-KAP ($\sigma = 0.1$) models on 500 samples from CIFAR10 dataset for varying amount of noise levels $\sigma$. Certified accuracy was estimated using the Monte Carlo procedure from (Cohen et al., 2019) using 100 samples for selection and 100000 samples for estimation.

### 4.4 KAP yields topographically organized kernels

As described in the methods, KAP combines the output of multiple kernels into a single activation passed from one layer to the next. As this operation creates dependencies between multiple kernels within each group, we expected a certain level of similarity between the kernels to emerge within each pseudo-ensemble. To examine this, we visualized the weights in the first convolutional layer of the normally trained (**NT**), adversarially trained (**AT**), and two variations of RN18 with Kernel Average Pooling (**KAP**) and Kernel Max Pooling (**KMP**) on CIFAR10 and Imagenet datasets (Fig. 7). As a reminder for these experiments, we had incorporated a 2-dimensional KAP that was applied on the kernels arranged on a 2-dimensional sheet.

As expected, the kernels in the **NT** and **AT** models did not show any topographical organization. Kernels in the **KMP** model were sparsely distributed, with many kernels containing very small weights. This was presumably because of the competition amongst different kernels within each pseudo-ensemble that had driven the network to ignore many of its kernels. In contrast, in the **KAP** model, we observed an overall topographical organization in the arrangement of the learned kernels. Moreover, the kernel weights often gradually shifted from the dominant pattern in one cluster to another (e.g. 45° angle edges to horizontal lines) as traversing along either of the two kernel dimensions. This topographical organization of the kernels on the 2-dimensional sheet is reminiscent of the topographical organization of the orientation selectivity of neurons in the primary visual cortex of primates (Hubel & Wiesel, 1977; Bosking et al., 1997).

| C10-NT | C10-KMP | C10-AT | C10-KAP | Imagenet-AT | Imagenet-KAP |

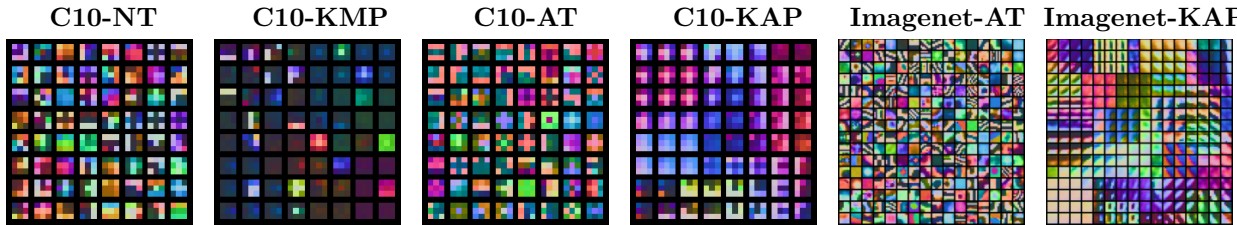

Figure 7: Visualization of the learned weights in the first layer of several variations of RN18 model.

## 5 Conclusion

We proposed Kernel Average Pooling as a mechanism for learning ensembles of kernels in layers of deep neural networks. We showed that when combined with activation noise, KAP-networks forge a process

that can be thought of as *recursive randomized smoothing with ensembles* applied at the level of features, where each stage consists in noise injection followed by applying ensemble of kernels. Our empirical results demonstrated significant improvement in network robustness at a fraction of computational cost of the state-of-the-art methods like adversarial training. However, because of the need for learning ensemble of kernels at each network layer, the improved robustness were often accompanied by reduced performance on the clean datasets. Future work should focus on addressing this downside of models with kernel ensembles. Nevertheless, our results suggest feature-level ensembling as a practical and scalable approach for training robust neural networks.

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

# A    Appendix

## A.1    2-Dimensionsal KAP

Given an input $\boldsymbol{z} \in \mathbb{R}^{D \times N_k}$, where $N_k = N_r \times N_c$ is the number of kernels that can be rearranged into $N_r$ rows and $N_c$ columns, and $D$ denotes the input dimension, the 2D kernel average pooling operation with kernel size $K \times K$ and stride $S$, computes the function

$$\bar{z}_{ik}(x) = \frac{1}{K^2} \sum_{l=\lfloor \frac{Sk}{N_c} \rfloor - \frac{K-1}{2}}^{\lfloor \frac{Sk}{N_c} \rfloor + \frac{K-1}{2}} \sum_{m=(Sk \bmod N_c) - \frac{K-1}{2}}^{(Sk \bmod N_c) + \frac{K-1}{2}} z_{i(lN_c+m)} \tag{10}$$

This procedure is graphically visualized in Fig. 1 and is further explained in Alg. 1, where the different operators are defined as such:

- The $Vec$ operator denotes the vectorization operation[1].

- The $Vec^{-1}_{D_1,D_2,D_3}$ denotes the inverse of the $Vec$ operator that reshapes a vector into a multi-dimensional tensor with a shape of $(D_1, D_2, D_3)$.

- The $Pad(x, p)$ operator denotes the 2D padding function applied on tensor $x$ that symmetrically appends $p$ zeros on both sides of the first two dimensions of the tensor $x$ with zero weight in computing the average.

- $AvePool(x, K, S)$ denotes the spatial average pooling operation applied on the first two dimensions of a tensor $x$ with kernel size $K$ and stride $S$.

---

**Algorithm 1** 2D Kernel Average Pooling

---

**Input:** layer input $\boldsymbol{x}$, kernel size $K$, stride $S$, vectorization operator $Vec$ and its inverse $Vec^{-1}$, zero padding function $Pad$, and average pooling function $AvePool$.
$\boldsymbol{z} \leftarrow Vec^{-1}_{\sqrt{D},\sqrt{D},WH}(Vec(\boldsymbol{x})^\top)$
$\boldsymbol{z} \leftarrow Pad(\boldsymbol{z}, \frac{K-1}{2})$
$\boldsymbol{z} \leftarrow AvePool(\boldsymbol{z}, K, S)$
$\boldsymbol{z} \leftarrow Vec^{-1}_{W,H,D}(Vec(\boldsymbol{z})^\top)$
**return:** $\boldsymbol{z}$

---

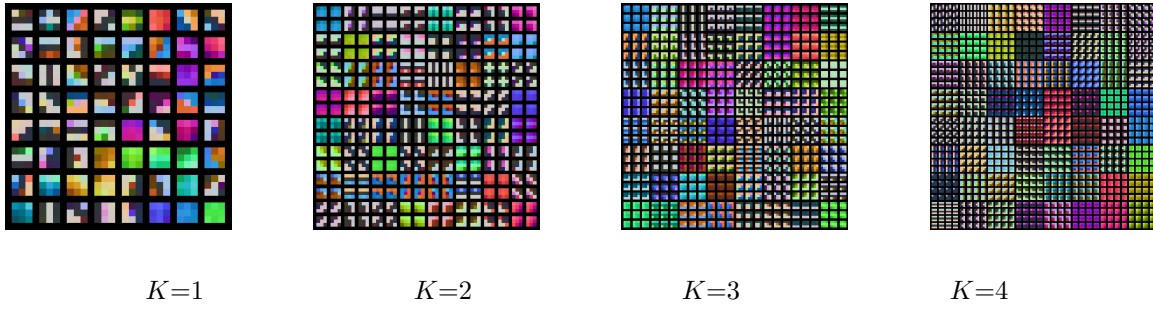

$K=1$ $\qquad\qquad$ $K=2$ $\qquad\qquad$ $K=3$ $\qquad\qquad$ $K=4$

Figure A1: Visualization of the learned weights in the first layer of 3-layer convolutional networks with varying KAP kernel size $K$.

---

[1]https://en.wikipedia.org/wiki/Vectorization_(mathematics)

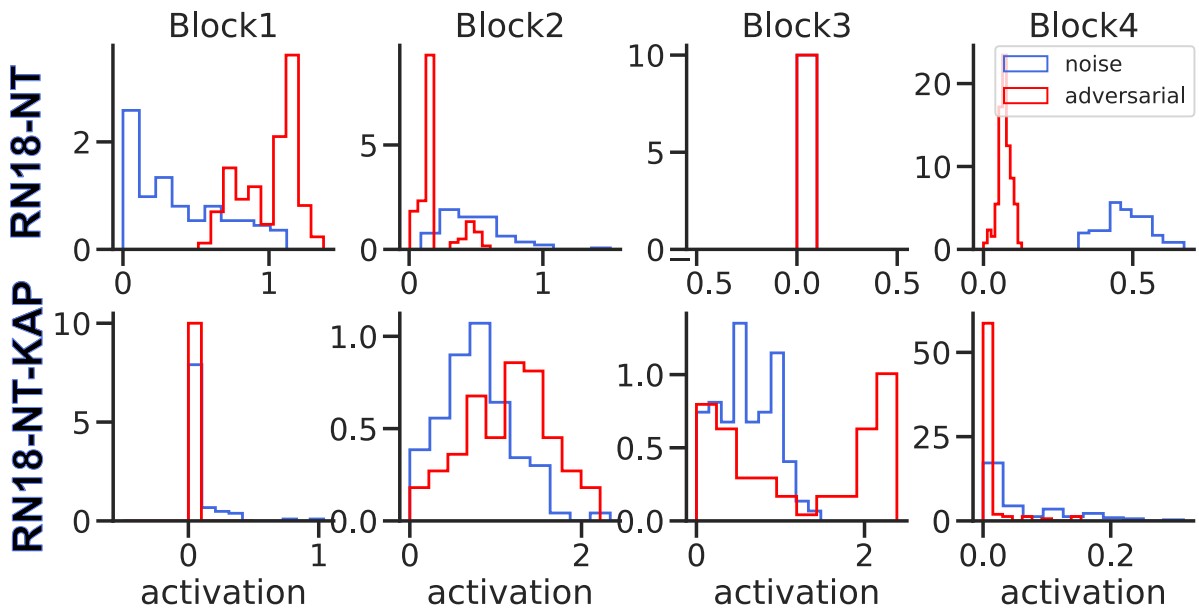

Figure A2: Distribution of layer activations in a random kernel for RN18-NT($\sigma = 0.1$) (top) and RN18-NT-KAP($\sigma = 0.1$) (bottom) trained on CIFAR10 dataset. We extracted the layer activations in response to a single random input image perturbed 100 times with random Gaussian noise or AutoAttack-$L_2$. Distributions of layer activations to noise and adversarial examples exhibit increased divergence in Block 4 in RN18-NT model but not in RN18-NT-KAP. The same pattern is replicated when considering other randomly selected input images from the validation set.

Table A1: Attack hyperparameters used to validate model robustness on each dataset.

| Attack | Dataset | Steps | Size ($\epsilon$) | More |
|---|---|---|---|---|
| PGD-$L_2$ | CIFAR TinyImagenet Imagenet | 100 | 1 | step=$\frac{4}{255}$ |
| AutoAttack | CIFAR TinyImagenet Imagenet | 100 | 1 | default standard AA - APGD-ce, APGD-t, FAB, SQUARE |
| SQUARE | CIFAR TinyImagenet Imagenet | 5000 | 1 | default SQUARE setting |

Table A2: Comparison of training and inference speed between alternative models. All training times were computed on the CIFAR10 dataset and ResNet18 architecture using a single A100 GPU.

| Dataset | Model | Ave. Training Time / Epoch | Ave. Inference Time / Batch |
|---|---|---|---|
| CIFAR10 | RN18-NT | $12.5 \pm 0.5$ sec | 9.43 msec |
| | RN18-AT | 100.75 sec | 9.43 msec |
| | RN18-KAP-NT ($\sigma = 0.1, K = 3$) | $14.5 \pm 0.5$ sec | 12.34 msec |

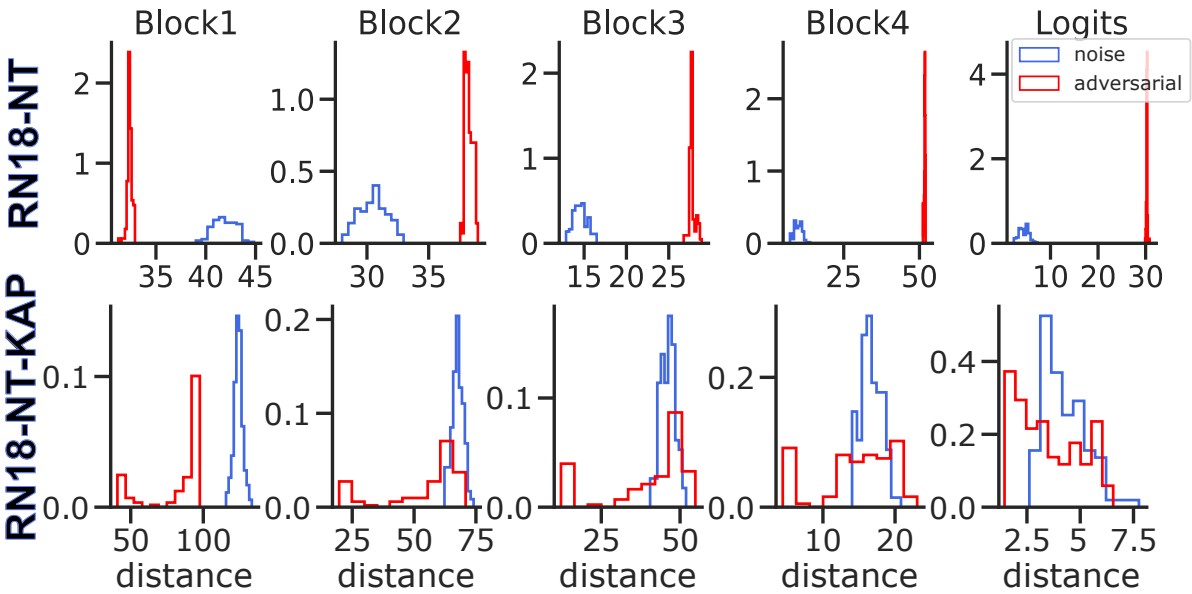

Figure A3: Distribution of perturbation magnitude in layer activations for RN18-NT($\sigma = 0.1$) (top) and RN18-NT-KAP($\sigma = 0.1$) (bottom) trained on CIFAR10 dataset. We extracted the layer activations in response to a single random input image perturbed 100 times with random Gaussian noise or AutoAttack-$L_2$. Perturbation magnitude is computed as the $L_2$ distance between perturbed and clean input activations ($f^{(l)}(x + n_i) - f^{(l)}(x)$ for noise and $f^{(l)}(x'_i) - f^{(l)}(x)$ for adversarial perturbations). Distributions of layer activations to noise and adversarial examples exhibits divergence in later blocks in RN18-NT model but not in RN18-NT-KAP.

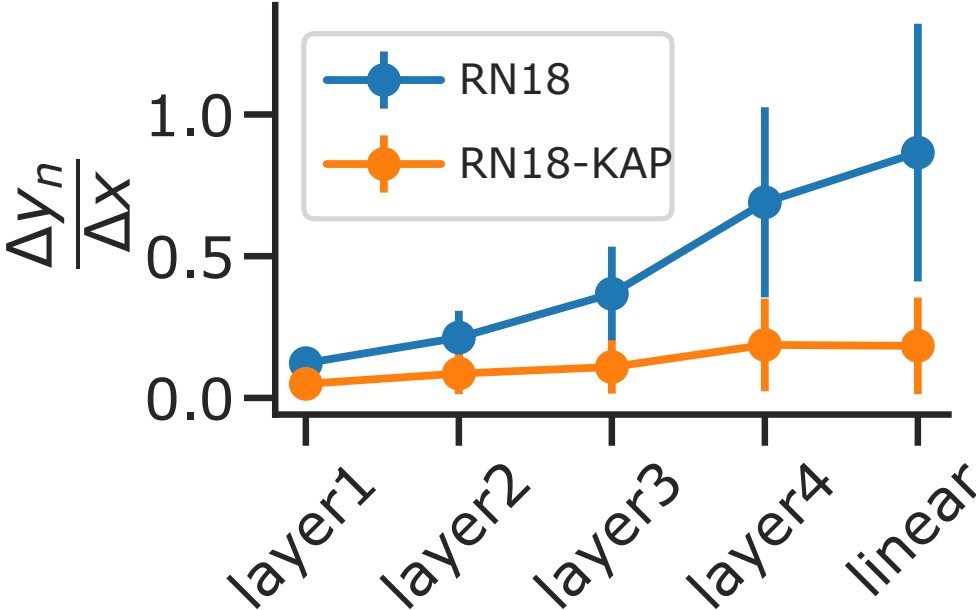

Figure A4: Normalized magnitude of change in each layer's activations in response to adversarial perturbations ($\frac{\|y_{adv} - y_{cln}\|}{\|y_{cln}\|}$) for AutoAttack-$L2, \epsilon = 1$. on CIFAR10 dataset for RN18-NT ($\sigma = 0.1$) and RN18-KAP-NT ($\sigma = 0.1$).

Table A3: Robust accuracy against transfer attacks on CIFAR10. Attacks were generated using each *Reference Model* and tested on the *Model*. We used AutoAttack-$L_2$ with $\epsilon = 1$.

| Model | Reference Model | AutoAttack |
|:---:|:---:|:---:|
| RN18-KAP-NT ($\sigma = 0.1, K = 3$) | RN18-NT | 77.1 |
| | RN18-AT | 65.4 |
| RN18-KAP-NT ($\sigma = 0.2, K = 3$) | RN18-NT | 71.8 |
| | RN18-AT | 62.0 |

