# OpenReview forum: "Learning Robust Kernel Ensembles with Kernel Average Pooling"
_TMLR — Rejected by TMLR_

### Review · Reviewer_2S7U · 2023-03-07

**Summary Of Contributions:**

This paper introduces Kernel Average Pooling (KAP) as a neural network building block that applies the mean filter along the kernel dimension of the layer activation tensor. KAP creates ensembles of kernels with similar functionality in convolutional neural networks, making them more robust to input perturbations. The authors propose using KAP on the internal layer of the network and considering additive Gaussian noise to improve adversarial robustness. They conduct empirical evaluations on various datasets.

**Audience:**

Yes

**Broader Impact Concerns:**

No ethical concerns.

**Claims And Evidence:**

No

**Requested Changes:**

I believe the overall evaluation and baseline reproduction should be updated. See the details above.

**Strengths And Weaknesses:**

**Strengths**\
The paper's related work and preliminary are well-considered, and the discussion on randomized smoothing is particularly noteworthy.

The author also thoughtfully considered various datasets for the evaluation.

**Weakness**\
The paper's evaluation has a critical flaw that makes it difficult to assess the validity of the contributions.

Firstly, if there is randomness present within the network, an evaluation utilizing EoT [4] would be more appropriate. For instance, AutoAttack [5] includes such an evaluation.

Secondly, the evaluation in the paper, except for AutoAttack, lacks meaning because AutoAttack is an ensemble of various attacks. Thus, an attack with higher accuracy than AutoAttack should not occur, at least in APGD, APGD-T, and Square, but this phenomenon exists in Tables 1 and 3, rendering the evaluation incorrect. Moreover, weak attacks should not be used for evaluation because they are defenses against worst-case scenarios [2].

Finally, the number of papers in Rice et al. [3] reported in Table 1 is too low. As demonstrated in [1], for AutoAttack, it should be at least 50.

**Overall**\
Given the evaluation's weaknesses, it is challenging to accept the paper's claims. Based solely on the results, it seems that the paper demonstrates robustness due to gradient masking [4]. The authors should consider a more rigorous adaptive evaluation [2] to address these issues.

**Reference**\
[1] Robust Overfitting may be mitigated by properly learned smoothening, ICLR 2021\
[2] On Adaptive Attacks to Adversarial Example Defenses, NeurIPS 2020\
[3] Overfitting in adversarially robust deep learning, ICML 2021\
[4] Obfuscated Gradients Give a False Sense of Security: Circumventing Defenses to Adversarial Examples, ICML 2018\
[5] Reliable evaluation of adversarial robustness with an ensemble of diverse parameter-free attacks, ICML 2020

---

> ### Author Response · Authors · 2023-03-21
> **Reliable robustness evaluations (part 1/2)**
>
> We thank the reviewer for providing feedback on our submission. The reviewer 1) questions the correctness of the reported baseline results and; 2) raises concerns with our robustness evaluations. In response, we argue that 1) our reported baseline results are correct and the reason for apparent differences between those in our work on other works cited by the reviewer is the differences in the specifications of the attacks used including the norm and $\epsilon$; 2) to address this issue we ran repeated evaluations for each model and reported the mean and std values for each of those tests. Our responses to specific comments are listed below. We believe our responses address the reviewer’s concerns and welcome any additional discussions.
>
> - *”if there is randomness present within the network, an evaluation utilizing EoT [4] would be more appropriate. For instance, AutoAttack [5] includes such an evaluation.”*
>
> Our model does not contain any randomness at test time. The activation noise is only added during training and not after.
>
> - *”the evaluation in the paper, except for AutoAttack, lacks meaning because AutoAttack is an ensemble of various attacks. Thus, an attack with higher accuracy than AutoAttack should not occur, at least in APGD, APGD-T, and Square, but this phenomenon exists in Tables 1 and 3, rendering the evaluation incorrect. Moreover, weak attacks should not be used for evaluation because they are defenses against worst-case scenarios [2].”*
>
> We agree with the reviewer that the strongest (i.e. worst case) attacks are more informative in terms of the model robustness and because of that reason our primary discussion of the results as well as all results presented in Figure 4 and 5 as well as results in Tables 1-3 are presented on AutoAttack results. On some datasets, the SQUARE accuray comes close to and slightly below that of AutoAttack. We reasoned that since SQUARE is already included in AutoAttack, the only way that these values may have been lower is because of the randomness in the attack itself. To get a more reliable evaluation, we repeated each of our evaluations 3 times and increased the sample size for Imagenet and TinyImagenet to 5000. The results in tables 1 and 2 are now updated.
>
> - *”the number of papers in Rice et al. [3] reported in Table 1 is too low. As demonstrated in [1], for AutoAttack, it should be at least 50.”*
>
> We suspect the reviewer is misremembering the results in Rice et al. and Chen et al. as there are several important differences between the results in those papers and ours. **1)** to the best of our knowledge, none of the results reported in those two papers are on AutoAttack and instead are on PGD; **2)** The majority of results reported in those papers were on networks trained using PGD $L_{\infty}$ with $\epsilon=\frac{8}{255}$ (all main figures and tables in Rice et al. except the ones reported the $L_2$ results in Table1). The only results that had used PGD $L_2$ in Table1 were carried out using $\epsilon=0.5$ which is half of the value used in our experiments ($\epsilon=1.$). Same differences apply to results in Chen et al. where they used $\epsilon=0.5$ in their PGD-$L_2$ evaluations. Since the reviewer mentions values close to 50, we suspect that the reviewer is referring to the $L_\infty$ results in those papers which should not be compared with the numbers reported in our work which use a different norm ($L_2$) and a different $\epsilon$ (1.0). As it has been shown in prior work, the differences in the norm and $\epsilon$ leads to substantial differences in the achieved robust errors [Maini et al. 2020, Bashivan et al. 2021].
>
> Maini, P., Wong, E. &amp; Kolter, Z.. (2020). Adversarial Robustness Against the Union of Multiple Perturbation Models. Proceedings of the 37th International Conference on Machine Learning.
> Bashivan, P., Bayat, R., Ibrahim, A., Ahuja, K., Faramarzi, M., Laleh, T., ... & Rish, I. (2021). Adversarial feature desensitization. Advances in Neural Information Processing Systems, 34, 10665-10677.
>
> [continued in the next comment]

---

> > ### Author Response · Authors · 2023-03-21
> > **Reliable robustness evaluations (part 2/2)**
> >
> > [continued from previous comment]
> >
> > - *“The authors should consider a more rigorous adaptive evaluation [2] to address these issues.”*
> >
> > We agree with the reviewer on the necessity of rigorous evaluations to assure robustness of any new proposed methodology. However, we believe that the reviewer’s comment is unspecific as “adaptive evaluation” is broadly used to refer to any evaluation of robustness that is tailored to a specific defense methodology. The paper referred by the reviewer discusses defense-specific attacks tailored to individual methods that broadly fit into several categories: a) use architectural motifs that are non-differentiable or mask gradients in one way or another; b) use objective terms that when included during attack, makes it difficult for the attacker to find good adversarial examples; c) use test-time stochasticity that makes gradient computation inaccurate; d) include non-standard or weak evaluations. Our proposed method does not fit into any of these categories as KAP only uses an averaging operation and does not include any non-differentiable components, does not use any custom objective functions, does not include test-time stochasticity, and is evaluated using a range of attacks including AutoAttack which is an ensemble attack consisting of both white-box and black box attacks that is widely accepted as one of the strongest attacks in the adversarial machine learning literature. In addition, we further tested the KAP models against adversarial examples transferred from adversarially and naturally trained models. We believe that these evaluations should already establish the effectiveness of our proposed method in improving robustness, however we remain open to any other suggestions by the reviewer.

---

### Review · Reviewer_ncMu · 2023-03-13

**Summary Of Contributions:**

This submission proposes kernel average pooling (KAP) to encourage neural networks to learn features, and improve adversarial robustness via random noise injection. Intuition and numerical analysis can been conducted to demonstrate the effectiveness of the proposed method.

**Audience:**

Yes

**Claims And Evidence:**

No

**Requested Changes:**

Please at least address my major concern about Allen-Zhu et al. (2021).

**Strengths And Weaknesses:**

Strength: This paper is clear and easy to understand.

Weakness: My major concern of this submission is the lack of evidence to support their claims. In particular, many of their intuitions and observations are somewhat contradict to the following paper:

Allen-Zhu, Zeyuan, and Yuanzhi Li. "Feature purification: How adversarial training performs robust deep learning." 2021 IEEE 62nd Annual Symposium on Foundations of Computer Science (FOCS). IEEE, 2022.

In Allen-Zhu et al. (2021), it is observed that clean training cannot purify hidden nodes and adversarial training is able to purify hidden nodes. They define "features" as the actual hidden factors of the data (assumed independent in their paper), and "purification" means each hidden node only learns one or a few features. They provide formal mathematical definitions of the terminologies and theoretical justifications to support their claims.

From my understanding, extending the logic behind Allen-Zhu et al. (2021), injecting noise during the training cannot help purify the hidden nodes, thus cannot improve adversarial robustness.

On the other hand, in terms of this submission,

[1] It is not clear what is the definition of "feature".

[2] From Figure 7, Imagenet, KAP learns a mixture of features in each node.

[3] The distributions of the activations for noise and adversarial attack in Figure A2 also seem to be different for me.

While introducing KAP leads to some changes in the training and robustness, the available evidences are somewhat contradict to the theory in Allen-Zhu et al. (2021), and are not strong enough to convince me.

===================================

Some other issues in this paper:

(1) In addition to the clean and robust accuracy performance of different methods, could the authors also provide the running time for each setting?

(2) For the result for RN18WideX4, the performance is different from what I would expect based on the results from

Xie, Cihang, and Alan Yuille. "Intriguing properties of adversarial training at scale." arXiv preprint arXiv:1906.03787 (2019).

The authors may want to compare the experimental settings in the above paper and this submission.

(3) For Figure 7, the range of the kernels are different for different methods, e.g., C10-KMP looks dark, and C10-AT has a color shift. I would suggest the authors to look into the range of the parameters in these methods, and try to provide a more comparable visualization for them.

---

> ### Author Response · Authors · 2023-03-21
> **Results are different but not contradictory (part 1/2)**
>
> We thank the reviewer for their valuable feedback. The reviewer raises concerns about seemingly contradicting results in our work compared to a recent paper by Allen-Zhu and Li (2021). Below, we discuss the assumptions underlying their theoretical work and the setup considered in their empirical work and highlight why their logic cannot be automatically extended to our setting and why our findings are different from theirs and not contradicting. Our responses to specific comments are listed below. We believe our responses address the reviewer’s concerns and welcome any additional discussions.
>
> - *”My major concern of this submission is the lack of evidence to support their claims. In particular, many of their intuitions and observations are somewhat contradict to the following paper: Allen-Zhu, Zeyuan, and Yuanzhi Li. "Feature purification: How adversarial training performs robust deep learning." 2021 [...] From my understanding, extending the logic behind Allen-Zhu et al. (2021), injecting noise during the training cannot help purify the hidden nodes, thus cannot improve adversarial robustness.”*
>
> We thank the reviewer for bringing the Allen-Zhu and Li (2021) paper to our attention. We believe that our intuitions and empirical observations are not in conflict with the claims of this paper.
>
> The theoretical work in Allen-Zhu and Li 2021 is done under strict assumptions on the data generation process and the classifier architecture. Specifically, their theoretical work assumes 1) a particular data generation process (Sparse Coding Model); 2) a particular neural network architecture (symmetric 2-layer ReLU network). Both of these assumptions are explicitly used in deriving their proofs and main claims about clean and adversarial training. Given that neither of these assumptions are necessarily true in our proposed evaluations, our findings do not contradict any of the theoretical findings in that paper.
>
> Moreover, their empirical results in section 8 also consider a different setup for clean training than ours. In particular, they perform vanilla clean training with no input augmentation or activation noise and in that regard their “clean training” is similar to the RN18-NT model that we have already considered as a baseline in our work. Importantly, this model is different from RN18-NT ($\sigma$=xx) which is trained with input and activation noise as well as RN18-KAP-NT models. In addition, the core of our proposal is the KAP operation which has not been considered in that paper. Because of these reasons, our results are not in conflict with the empirical findings of that paper either.
>
> Finally, there is an expanding literature that empirically (Ford et al. 2019, Dapello et al 2021) and theoretically (Cohen et al. 2019) demonstrates the effectiveness of training on noise-corrupted inputs and activations on improving robustness to adversarial perturbations in neural networks. We have reviewed many of these publications in length in our “related works and background” section and invite the reviewer to revisit that section.
>
> Joel Dapello, Jenelle Feather, Hang Le, Tiago Marques, David D. Cox, Josh H. McDermott, James J. DiCarlo, and SueYeon Chung. Neural Population Geometry Reveals the Role of Stochasticity in Robust Perception. In NeurIPS, pp. 1–13, 2021
>
> Nicolas Ford, Justin Gilmer, Nicholas Carlini, and Ekin D. Cubuk. Adversarial examples are a natural consequence of test error in noise. In 36th International Conference on Machine Learning, ICML 2019, volume 2019-June, pp. 4115–4139, 2019.
>
> Jeremy Cohen, Elan Rosenfeld, and J. Zico Kolter. Certified adversarial robustness via randomized smooth-ing. 36th International Conference on Machine Learning, ICML 2019, 2019-June:2323–2356, 2019.
>
>
> - *”It is not clear what is the definition of "feature".”*
>
> In our work, we have used the term “feature”, to refer to the unit activity at the intermediate layers of the network. We now clarify this definition in the introduction.
>
> - *”The distributions of the activations for noise and adversarial attack in Figure A2 also seem to be different for me.”*
>
> Comparing the top row (RN18-NT) and bottom (RN18-KAP-NT), we hope that it is evident to the reviewer that the distributions of activations in RN18-KAP-NT are much more overlapping between the noise and adversarial samples compared to RN18-NT.
>
> - *”In addition to the clean and robust accuracy performance of different methods, could the authors also provide the running time for each setting?”*
>
> We are unsure whether the reviewer is asking for the running time during training or at inference (test) time. We had already included the training time comparisons between different methods in Table-A2. We further included a comparison of inference time between the methods and included them in the same table.
>
> [continued in the next comment]

---

> > ### Author Response · Authors · 2023-03-21
> > **Results are different but not contradictory (part 2/2)**
> >
> > [continued from previous comment]
> >
> > - *”For the result for RN18WideX4, the performance is different from what I would expect based on the results from Xie, Cihang, and Alan Yuille. [...] The authors may want to compare the experimental settings in the above paper and this submission.”*
> >
> > There are two major differences between our reported results and those in Cihang and Yuille 2019 paper: 1) all networks considered in that work have substantially deeper architectures. The shallowest network considered in that work is ResNet-152 which much deeper than ResNet-18 architecture considered in our work; 2) unlike that work which uses the vanilla adversarial training, because of the high computational cost of adversarial training on the Imagenet dataset, as mentioned in section 4.1, we used the fast adversarial training (Wong et al 2019) to train the AT network on Imagenet. We believe that because of these differences, the numbers reported in that work and ours are not comparable. As demonstrated in Wong et al 2019, the results of the Fast-AT method are very close to that achieved by adversarial training.
> >
> > Eric Wong, Leslie Rice, and J Zico Kolter. Fast is better than free: Revisiting adversarial training. In International Conference on Learning Representations, 2019.
> >
> > - *”For Figure 7, the range of the kernels are different for different methods, e.g., C10-KMP looks dark, and C10-AT has a color shift. I would suggest the authors to look into the range of the parameters in these methods, and try to provide a more comparable visualization for them.”*
> >
> > For visualizing the weights, we use a standard visualization tool from pyTorch (torchvision.utils.make_grid) with normalization set to True. Normalization computes the min and max values across all weights and normalizes all weights to map to 0-1 range before visualization. This avoids assigning separate value ranges to different kernels and thus in our opinion a more informative way for visualizing the weights.

---

> > ### Comment · Reviewer_ncMu · 2023-03-21
> > **About Allen-Zhu et al. 2021**
> >
> > I appreciate the authors' feedback towards my comments. Most of my concerns are address.
> >
> > I'm still concerning about Allen-Zhu et al. (2021). If the authors believe there is no conflict, please at least provide some characteristics of data (not just numerical experiment) where KAP is proven to improve adversarial robustness via noise injection.

---

> > > ### Author Response · Authors · 2023-03-22
> > > **Clarification**
> > >
> > > We thank the reviewer for the additional comment. However, we are unsure about what the reviewer is precisely asking. We would appreciate it if the reviewer would 1) let us know which part of the arguments made in our response was not convincing and 2) which form of “characteristics of data” they have in mind. The answer to these two questions will immensely help us in providing a better response to your comment.

---

### Review · Reviewer_hiro · 2023-03-15

**Summary Of Contributions:**

This paper studies the problem of adversarial defense and proposes a Kernel Average Pooling (KAP) operation along with random noise injection to improve the adversarial robustness of deep neural networks (DNNs). The proposed KAP operation squeezes (averages out) the internal activation along the channel dimension to create an "ensemble" of kernels (or subnets) to improve the robustness to small input perturbations. Contributions include:
1. The study of internal robustness of DNNs.
2. Proposed a new operation to create kernel ensembles.

**Audience:**

Yes

**Broader Impact Concerns:**

No concerns.

**Claims And Evidence:**

No

**Requested Changes:**

See the above Weaknesses.

**Strengths And Weaknesses:**

Strengths:

1. An efficient way to create kernel "ensemble" to improve the robustness of DNNs

Weaknesses:
1. Lack of knowledge of the field of adversarial machine learning. Adversarial examples were first discovered by Szegedy et al. [1] and Biggio et al. [2], yet neither work was cited in this paper.
2. In Section 3.2, it is confusing how the KAP is defined, $S$ is a stride hyperparameter, and why it appears in Eq. (2) and (3)?
3. Confusing notations in Section 3.3., for example, 1) the so-called "two sets of weights" are actually weights of the two layers of the network; 2) the example network in Figure 2 is actually a two-layer neural network, not one layer neural network; 3) $w_i^{(1)}$ is in bold while $w_1^{(2)}$ is not.
4. Figure 1 is also not clear why after the average pooling the size stays the same.
5.  The proposed method is a channel averaging operation without learnable parameters, which is similar to [3] and [4]. But no learnable parameters and the averaging operation do not impact the calculation of the gradients in the tensor graph, not the case computed in Eq. (6). It is very likely that the "small robustness" comes from the randomized smoothing rather than the proposed KAP.
6. In Table 1, the results against the AutoAttack under standard epsilon indicate that there is almost no robustness improvement. Also, standard adversarial training methods like Madry's and TRADES should be compared as baselines.

[1] Szegedy, Christian, et al. "Intriguing properties of neural networks." arXiv preprint arXiv:1312.6199 (2013).
[2] Biggio, Battista, et al. "Evasion attacks against machine learning at test time." Machine Learning and Knowledge Discovery in Databases: European Conference, ECML PKDD 2013, Prague, Czech Republic, September 23-27, 2013, Proceedings, Part III 13. Springer Berlin Heidelberg, 2013.
[3] Bai, Yang, et al. "Improving adversarial robustness via channel-wise activation suppressing." arXiv preprint arXiv:2103.08307 (2021).
[4] Yan, Hanshu, et al. "Cifs: Improving adversarial robustness of cnns via channel-wise importance-based feature selection." International Conference on Machine Learning. PMLR, 2021.

---

> ### Author Response · Authors · 2023-03-21
> **Clarifications on methodology and highlighting the results (part 1/2)**
>
> We thank the reviewer for providing feedback on our submission. The reviewer 1) comments on the clarity of our methodology; 2) raises issues with the effectiveness of our proposed method and; 3) suggests additional baseline experiments. In response, 1) we clarify the points raised by the reviewer and revise the manuscript to reflect those potential confusion points; 2) we argue that our proposed method demonstrates substantial improvements to network robustness at a fraction of computational cost to most existing approaches to adversarial robustness. We further argue that these improvements to network robustness are not entirely because of training on additive Gaussian noise and that the Kernel Average Pooling operation has a critical role in that; 3) we add the TRADES baseline to the results reported in Table-1. Our detailed responses to specific comments are listed below. We believe our responses address the reviewer’s concerns and welcome any additional discussions.
>
> - *”Lack of knowledge of the field of adversarial machine learning. Adversarial examples were first discovered by Szegedy et al. [1] and Biggio et al. [2], yet neither work was cited in this paper.”*
>
> We thank the reviewer for pointing out the two papers. We had by mistake referred to another paper by Goodfellow, Shlens and Szegedy on adversarial examples. We have now addressed this issue and added the two suggested references.
>
> - *”In Section 3.2, it is confusing how the KAP is defined, S  is a stride hyperparameter, and why it appears in Eq. (2) and (3)?”*
>
> As mentioned by the reviewer, $S$ is the stride parameter. Thus, similar to spatial average pooling, $S$ and $K$ (kernel size) determine the range of inputs $z_i$ that are averaged to compute each output $i$. As an example, when K=3 and S=2: $z_2 = \frac{1}{3}\sum_{3}^{5}z_l$ which demonstrates how the range of inputs depend on both $S$ and $K$ parameters.
>
> - *”Confusing notations in Section 3.3., for example, 1) the so-called "two sets of weights" are actually weights of the two layers of the network; 2) the example network in Figure 2 is actually a two-layer neural network, not one layer neural network; 3) wi(1) is in bold while w1(2) is not.”*
>
> To answer the reviewer’s comment: 1) $W^{(1)}$ and $\boldsymbol{w}^{(2)}$ correspond to the connection weights between the input and hidden layer, and hidden layer to output layer respectively. This relation is visually depicted in Figure 2 and is referred to at the second line of section 3.3; 2) we thank the reviewer for pointing out this error. The term “one-layer neural net” used in Figure 2 caption is incorrect and should be “neural network with one hidden layer” similar to how this network is referred to in section 3.3 text. We have addressed this issue by revising Figure 2 caption to match the main text; 3) Following the notations used in the Deep Learning Book by Goodfellow, Bengio, and Courville, we use bold symbols for vectors and regular symbols for scalar values. In this case, $\boldsymbol{w}_i^{(1)}$ is a vector and $w_i^{(2)}$ is a scalar.
>
> - *”Figure 1 is also not clear why after the average pooling the size stays the same.”*
>
> As explained in section 3.2 (under Eq. 2), we zero-pad the inputs to match the dimensionality of the KAP input and output. To make this point more clear in the figure, we also added this point in Figure 1 caption.
>
> - *“The proposed method is a channel averaging operation without learnable parameters, which is similar to [3] and [4]. But no learnable parameters and the averaging operation do not impact the calculation of the gradients in the tensor graph, not the case computed in Eq. (6).”*
>
> We thank the reviewer for bringing the two additional references to our attention. As mentioned by the reviewer, these two papers discuss completely different ideas than that in our work and propose methods for regulating channel activity in the context of adversarial training. We have added these two papers to our literature review section.
>
> **We are unsure what the reviewer means by “the averaging operation do not impact the calculation of the gradients in the tensor graph, not the case computed in Eq. (6)” and would appreciate it if the reviewer could further clarify what they mean by it.**
>
> [Continued in next comment]

---

> > ### Author Response · Authors · 2023-03-21
> > **Clarifications on methodology and highlighting the results (part 2/2)**
> >
> > [continued from previous comment]
> >
> > - *“It is very likely that the "small robustness" comes from the randomized smoothing rather than the proposed KAP.” and “In Table 1, the results against the AutoAttack under standard epsilon indicate that there is almost no robustness improvement.”*
> >
> > We suspect that the reviewer may have misread or misunderstood our results and that is why they refer to the improvements by our method as "small robustness". Our empirical results on four different datasets and different neural network architectures all clearly demonstrate ***large gains in adversarial robustness*** using our approach. Accuracy against AutoAttack on CIFAR10 increases from 0 to ~31% (Table1-top), on CIFAR100 from 0 to ~12% (Table1-bottom),  on TinyImagenet to ~16% (Table2-top), on Imagenet to ~27% which is higher than the model trained with fast adversarial training (Table2-bottom). Importantly, these gains in network robustness are obtained without the computationally costly adversarial training procedure and only by training the network on random Gaussian noise.
> >
> >
> > We also specifically demonstrated the contribution of KAP and training on additive Gaussian noise and highlighted the importance of both components. In Figure 4a, we compare the robust accuracy of CNNs with and without KAP and show that network architectures with KAP substantially outperform those without it. We further demonstrate this point in Figure 5a in which we compare the robust accuracy in ResNet18 and ResNet18-KAP where each convolutional layer is followed by a KAP operation and again confirming that KAP boosts the robust accuracy even without training on additive Gaussian noise.
> >
> >
> > - *“Also, standard adversarial training methods like Madry's and TRADES should be compared as baselines.”*
> >
> > In our results we have already included the results for the adversarial training method with early stopping (Rice et al. 2020) which as demonstrated in that paper is substantially higher than that originally proposed by Madry et al. Following the reviewer’s suggestion, we also included the results from TRADES as an additional baseline in Table1.

---

### Public Comment · ~Paul_Gavrikov1 · 2023-03-29
**Related Work**

The authors mention the work of (Wang et al., 2020): " showed that neural network models which their[sic!] weights are encouraged to be smooth during training yield improved adversarial robustness."
From what I understand, this is not true. Wang shows that robust models tend to have smoother kernels, and, that smoothing **after** training gains robustness. They do not regularize.

There's also some other work by me into robust kernels which may be relevant:

Paul Gavrikov, Janis Keuper "CNN Filter DB: An Empirical Investigation of Trained Convolutional Filters", Proceedings of the IEEE/CVF Conference on Computer Vision and Pattern Recognition (CVPR), 2022, pp. 19066-19076

Paul Gavrikov, Janis Keuper, "Adversarial Robustness Through the Lens of Convolutional Filters", Proceedings of the IEEE/CVF Conference on Computer Vision and Pattern Recognition (CVPR) Workshops, 2022, pp. 139-147

---

> ### Author Response · Authors · 2023-04-07
> **Thank you**
>
> Thank you for the comment. We will update the text in the next revision of the manuscript.

---

### Decision · Action_Editors · 2023-04-21

**Recommendation:** Reject

**Comment:**

This paper introduces Kernel Average Pooling (KAP) as a neural network building block that applies the mean filter along the kernel dimension of the layer activation tensor. The authors show that the KAP operation along with random noise injection can improve the adversarial robustness of deep neural networks (DNNs).

After author response, this paper received Reject, Leaning Reject, and Leaning Accept recommendations. On one hand, the proposed idea in this paper is interesting. One reviewer also commented that there was some critical misunderstandings in the initial review regarding the experimental evaluations, and the author response properly addressed the concern. On the other hand, the other two reviewers still showed major concerns about the paper, commenting that a more systematic evaluation and comparison with existing works like randomized smoothing and adversarial training is needed, and the authors did not convincingly show the significance of the proposed method, especially when compared with many existing techniques like mean filter, smoothing, suppressing, (fast) adversarial training, etc. The empirical studies are not strong enough to support their claims.

The largest experiment setting in this paper is using ResNet18-WideX4 with ImageNet. In terms of this, the editor also had some concerns regarding the experiment scope, and testing the proposed method with stronger ResNet backbones will make the results stronger, but surely more computation resources are required here. Also, it seems unclear whether the proposed method can be generalized to vision transformer or not, as ViT is a popular backbone choice for vision tasks nowadays.

Given the overall (slightly) negative review comments, on balance, the editor thinks that the flaws slightly outweigh the merits, and would like to recommend rejection of the paper by the end. The editor however would be willing to consider a major revision that would address the points mentioned above, following another round of reviews.

**Audience:**

Yes.

**Claims And Evidence:**

The claims made in the submission may be partially supported by clear and convincing evidence, as shown in some of the concerns listed by the reviewers.